# Sex-dimorphic expression of extracellular matrix genes in mouse bone marrow neutrophils

Cassandra J. McGill[1], Collin Y. Ewald[2]*, Bérénice A. Benayoun[1,3,4,5,6]*

**1** Leonard Davis School of Gerontology, University of Southern California, Los Angeles, California, United States of America, **2** Laboratory of Extracellular Matrix Regeneration, Department of Health Sciences and Technology, Institute of Translational Medicine, Swiss Federal Institute of Technology (ETH Zürich), Schwerzenbach, Switzerland, **3** Molecular and Computational Biology Department, USC Dornsife College of Letters, Arts and Sciences, Los Angeles, California, United States of America, **4** Biochemistry and Molecular Medicine Department, USC Keck School of Medicine, Los Angeles, California, United States of America, **5** USC Norris Comprehensive Cancer Center, Epigenetics and Gene Regulation, Los Angeles, California, United States of America, **6** USC Stem Cell Initiative, Los Angeles, California, United States of America

* berenice.benayoun@usc.edu (BAB); collin-ewald@ethz.ch (CYE)

**Data Availability Statement:** The bulk RNA-seq data was previously described in Lu et al., 2021 [11], and sequencing data is accessible through BioProject PRJNA630663 (https://www.ncbi.nlm.nih.gov/sra?LinkName=bioproject_sra_all&from_

## Abstract

The mammalian innate immune system is sex-dimorphic. Neutrophils are the most abundant leukocyte in humans and represent innate immunity's first line of defense. We previously found that primary mouse bone marrow neutrophils show widespread sex-dimorphism throughout life, including at the transcriptional level. Extracellular matrix [ECM]-related terms were observed among the top sex-dimorphic genes. Since the ECM is emerging as an important regulator of innate immune responses, we sought to further investigate the transcriptomic profile of primary mouse bone marrow neutrophils at both the bulk and single-cell level to understand how biological sex may influence ECM component expression in neutrophils throughout life. Here, using curated gene lists of ECM components and unbiased weighted gene co-expression network analysis [WGCNA], we find that multiple ECM-related gene sets show widespread female-bias in expression in primary mouse neutrophils. Since many immune-related diseases (*e.g.*, rheumatoid arthritis) are more prevalent in females, our work may provide insights into the pathogenesis of sex-dimorphic inflammatory diseases.

## Introduction

Accumulating evidence shows that, even outside of reproductive organs, mammalian biology is very sex-dimorphic [1]. Indeed, the mammalian immune system shows broad differences between males vs. females, including aspects of innate and adaptive immune responses [2–4]. Generally, a more robust immune response is observed in females, which may underlie a higher auto-immunity risk, whereas an increased susceptibility to infection and a worse response to sepsis is found in males [2].

Neutrophils are the most abundant white blood cell type in human blood, representing 50–70% of leukocytes in humans throughout life [5]. They are a type of "granulocytes", produced

uid=630663). The processed normalized count table and DEseq2 result tables were obtained from Github (https://github.com/BenayounLaboratory/Neutrophil_Omics_2020). The single-cell RNA-seq data was previously described in Kim et al., 2022 [12], and raw sequencing data is accessible through BioProject PRJNA796634 (https://www.ncbi.nlm.nih.gov/sra?LinkName=bioproject_sra_all&from_uid=796634). The processed annotated Seurat file was obtained from Figshare (https://doi.org/10.6084/m9.figshare.19623978). The serum proteomics dataset was previously described in Aumailley et al, 2021 [22], and the processed protein expression matrix was obtained from the online Table S2 for the article (https://pubs.acs.org/doi/suppl/10.1021/acs.jproteome.1c00542/suppl_file/pr1c00542_si_006.xlsx).

**Funding:** This work was supported by National Institute of Aging [https://www.nia.nih.gov/] T32 AG052374 predoctoral fellowship to C.J.M., by the Swiss National Science Foundation [https://www.snf.ch/en] from the SNF P3 Project 190072 to CYE, National Institute of Aging [https://www.nia.nih.gov/] R01 AG076433 to B.A.B. and Pew Biomedical Scholar award #00034120 from the Pew Charitable Trust [https://www.pewtrusts.org/] to B.A.B. The funders had no role in study design, data collection and analysis, decision to publish, or preparation of the manuscript.

**Competing interests:** The authors have declared that no competing interests exist.

in the bone marrow and released into circulation for immune surveillance, representing a first line-of-defense against infections [6, 7]. Neutrophils perform key immune functions, including the production/release of antimicrobial granules and of "Neutrophil Extracellular Traps" (NETs) [6, 8], composed of extruded chromatin, proteases, and antimicrobial peptides. While neutrophils defend against infections, their aberrant activation aggravates pathological inflammation [6, 7, 9]. They can also drive inflammatory disease, leading to tissue damage [8, 10].

Recent work by our group revealed widespread sex-dimorphism in the transcriptome of primary murine bone marrow neutrophils throughout life (4 to 20 months of age) [11]. Consistently, transcriptomic data by the ImmGen Consortium showed clear differences between the neutrophils of young prepubertal female *vs.* male mice (6 weeks of age) [4]. Importantly, we also recently reported that sex-differences in primary bone marrow neutrophils are found at every stage of underlying heterogeneity using single-cell RNA-seq [12]. In our previous analyses of primary neutrophil transcriptomes, Gene Ontology [GO] terms related to collagen synthesis and extracellular matrix (ECM) biology were observed in the top listed GO terms with significant sex-dimorphic gene expression at the bulk RNA-seq level [11]. Furthermore, beyond the transcriptional level, accumulating evidence supports the idea that the functional landscape of neutrophils differs between sexes throughout life, with male-bias in the production of primary granule protein neutrophil elastase (ELANE) [11], a serine protease that can break down elastin and collagens [13], potentially remodels the ECM.

Intriguingly, ECM components are emerging as potent regulatory mediators of innate immune responses [14]. The importance of the extracellular matrix in neutrophil recruitment and function is becoming more evident based on recent studies [15]. Indeed, recent work suggests that neutrophils can transport ECM components to wound sites to help promote healing [16]. Although the production of ECM components by neutrophils themselves has not been explored in depth, emerging evidence suggests that the synthesis of "emergency" ECM containing fibronectins by neutrophils plays an important role in promoting fracture healing [17]. However, sex-differences in the regulation of ECM biology by neutrophils remain poorly understood.

To follow-up on this observation of sex-dimorphism in ECM gene expression, we reanalyze transcriptomic data from primary mouse bone marrow neutrophils to help understand how biological sex may influence ECM component gene expression. Using expert-curated ECM gene sets, we confirm broad female-bias in the expression of genes related to ECM components and remodeling in young and old female neutrophils in our bulk RNA-seq dataset. To gain a more granular view and determine any underlying heterogeneity of ECM expression of young mice's neutrophils, we recently generated a single cell RNA-seq data and find again a female-specific enrichment for ECM. In addition, using unbiased weighted gene co-expression network analysis (WGCNA), we identify significant sex-biased gene expression modules in primary neutrophils. Importantly, the top female-biased WGCNA module in neutrophils was significantly enriched for ECM-related genes. Thus, our analyses suggest that female neutrophils may uniquely contribute to tissue repair and ECM remodeling, in addition to their core function in the immune response.

## Results

### Gene set enrichment analysis confirms female-biased expression of ECM-related genes expression in primary mouse neutrophils

In our previous study using a multi-omic approach to assess changes in neutrophils based on age and sex, we observed a significant female bias (regardless of age) for expression of genes related to GO terms "Collagen chain trimerization", "Assembly of collagen fibrils and other

multimeric structures", and "Non-integrin membrane-ECM interactions" [11]. Because of the emerging role of neutrophils in ECM-remodeling [15], we decided to investigate further the potential differential expression of extracellular matrix related processes, as a function of organismal aging or biological sex.

To determine whether ECM-related genes are differentially regulated as a function of organismal age or biological sex, we took advantage of 2 datasets that we previously generated: (i) a bulk transcriptomic dataset of primary bone marrow neutrophils derived from young and old, female and male C57BL/6Nia mice [11], and (ii) a single cell transcriptomic dataset of primary bone marrow neutrophils derived from young female and male C57BL/6J mice [12] (Fig 1A). In addition, to maximize our understanding of the differential expression of ECM-related genes in neutrophils, we leveraged expert-curated gene sets related to ECM biology (hereafter matrisome; S1 Table). The matrisome encompasses all proteins that form the extracellular matrix (i.e., collagens, ECM glycoproteins, proteoglycans), also referred to as the core matrisome, encompassing 274 genes in mice [18]. Furthermore, the matrisome also includes proteins that either remodel the ECM (i.e., matrix metalloproteases or ECM regulators) or localize into the ECM (i.e., secreted factors, ECM-affiliated genes), collectively referred to as matrisome-associated genes, which are 836 genes in mice [18]. Thus, examining the changes in the composition of these 1,110 matrisome genes might help us reveal some new underlying biology.

First, we leveraged these gene sets to perform Gene Set Enrichment Analysis (GSEA) on bulk transcriptomic neutrophil data to detect potential changes as a function of aging or sex (Fig 1B; S2A and S2B Table). We observed no significant changes in the expression of ECM-related genes based on organismal age, suggesting that ECM biology is not broadly disrupted in aging mouse neutrophils (S2A Table). By contrast, we did find robust female bias in the expression of most ECM-related gene sets in neutrophils (Fig 1B; S2B Table). In addition to a female-biased enrichment in collagen-encoding genes (Fig 1B, FDR = $4.26 \times 10^{-9}$), we also observed significant female-biased expression for genes related to the core matrisome, ECM affiliated genes, ECM glycoproteins, matrisome-associated genes, and secreted factors, but not for proteoglycan-related genes (Fig 1B; S2B Table). Using Fisher's exact test for the significance of overlap, we confirmed that genes from ECM-related gene sets tended to be overrepresented among genes with significant female-biased neutrophil expression by bulk RNA-sequencing (Genes biased for female expression according to DESeq2 at FDR < 5%; S1A Fig; S2C Table). Interestingly, genes expressed at significantly higher levels in female neutrophils show 1.7- to 7.9-fold enrichment for ECM-related genes compared to chance (S2C Table).

We were also curious whether the female ECM-related gene expression bias would also be observed at the single-cell level. For this purpose, we took advantage of a recent single-cell RNA-seq dataset from female vs. male mouse bone marrow neutrophils [12]. To minimize the impact of "dropouts" in single-cell RNA-seq datasets [19], we used a gene set scoring approach, UCell [20], to determine the overall expression of the ECM-related gene set in each individual cell. This approach computes a single numeric value per cell based on the expression of genes in a gene set of interest without the need for expression thresholding, thus increasing statistical power [20]. We show the summarized UCell scoring values for each ECM-related gene set per biological sample, and the significance of the difference in score distributions in female vs. male neutrophils (Fig 1C). Importantly, this analysis revealed a robust female-bias expression for most of the ECM-related gene sets (Fig 1C). Interestingly, based on neutrophil subset definitions from Xie et al. (2020) [21], ECM-related genes may be highest in more mature neutrophils, suggesting that this phenotype is acquired during the maturation process (S1B Fig).

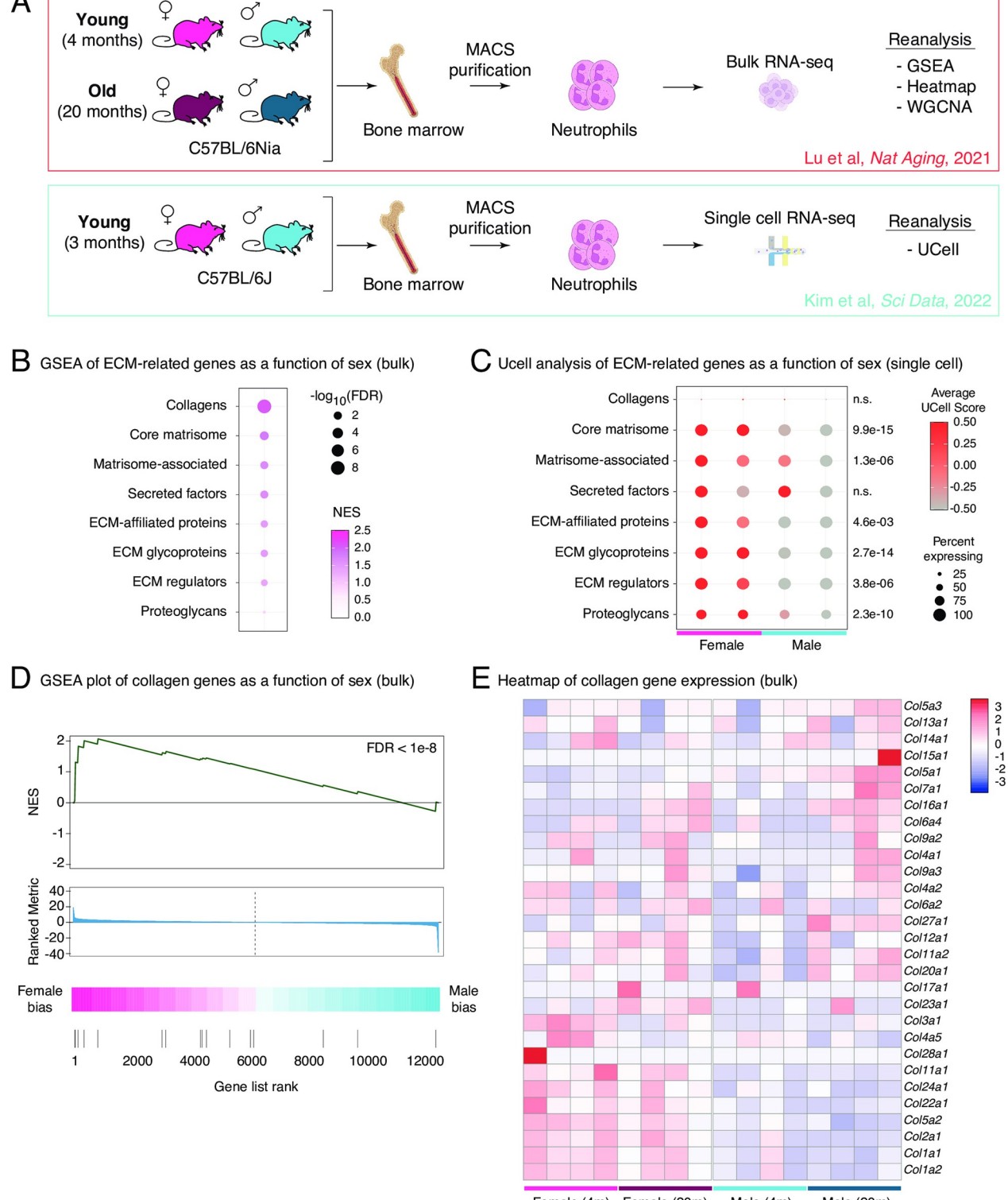

**Fig 1. Female neutrophils exhibit higher transcriptional expression of extra-cellular matrix related genes.** (**A**) Experimental design of primary neutrophil transcriptomic datasets reanalyzed in this study [11, 12]. Panel created with BioRender.com. (**B**) Bubble plot showing GSEA enrichment results for ECM-related genes based on sex expression from neutrophil bulk RNA-seq (all terms are female-biased). See full analysis results in **S2B Table**. (**C**) Dotplot of 'Ucell' scores of ECM-related genesets aggregated by sample of origin in single cell neutrophil RNA-seq dataset from [12]. For each geneset, statistical significance in a non-parametric Wilcoxon rank sum test between the distribution of UCell scores between female and male

neutrophils is reported. (**D**) GSEA plot of collagen genes from neutrophil bulk RNA-seq (FDR < 1e-8, indicative of female neutrophil expression bias). (**E**) Heatmap displaying the expression of collagen genes (gene set identical as in C) from the bulk neutrophil RNA-seq.

Interestingly, although collagens were not robustly detected in single-cell RNA-seq due to relatively low expression and sensitivity of the assay (<10% of cells showed detectable collagen transcript levels in single-cell RNA-seq regardless of sex, which makes single-cell level analysis not robust for this class of genes; **Fig 1C**), collagen genes showed robust female-bias in their expression at the bulk transcriptome level (**Fig 1D and 1E**).

Although we could not find a publicly available mouse neutrophil quantitative proteomics dataset with female and male animals, we identified a high-quality serum proteomics dataset derived from young C57BL/6NHsd female and male mice [22]. Since neutrophils represent a large proportion of circulating blood leukocytes [5], we reasoned that differences in the production of ECM-related proteins by neutrophils should be at least partially reflected in the serum proteome. Thus, we identified significantly sex-biased serum proteins using Limma [23], with respectively 73 female-biased and 42 male-biased serum proteins at FDR < 5% (**S1C and S1D Fig**; **S2D Table**). Consistent with our transcriptomics observations, ECM-related gene sets tend to be enriched among serum female-biased proteins (Limma FDR < 5%; **S1E Fig**; **S2E Table**). While neutrophils are likely contributing to this sex-bias in serum ECM proteomics signatures, this observation may also indicate a more general sex bias in ECM expression among immune/circulating cells.

Overall, our analysis determined that genes that both encode structural components of the ECM and remodel the ECM tend to be expressed at higher levels in female neutrophils throughout life (and found at higher protein levels in the serum), suggesting that the emerging interactions between neutrophil activity and ECM biology may be sex-dimorphic throughout life.

## Weighted gene co-expression network analysis reveals that the top female-biased module in neutrophil transcriptomes is enriched for collagens

We next asked which gene modules were coregulated together with ECM-related genes in a female-biased fashion by leveraging WGCNA (**S2A Fig**). Since WCGNA is optimized to be used on bulk datasets, we focused this analysis on the bulk transcriptomic neutrophil data. After network construction, we identified 13 significant transcriptional modules (**S3 Table**). We next used the WGCNA module-trait relationship feature to identify which modules were significantly associated with organismal age and/or sex (**Fig 2A**).

Most notably, the most significant sex-associated module, the Salmon module, which contains 318 genes, showed clear female-bias regardless of age (**Fig 2B**, **S2B Fig**). To note, there was no significant enrichment for X-linked genes in the Salmon module (p = 0.9153 in Fisher's exact test), although some X-linked genes were found in this module (*i.e.*, *Xist*, *Tsix*, *Nyx*, *Kdm6a*, *Kdm5c*, *Zxdb)*. Manual inspection also revealed that this module contains many collagen genes (**S3 Table**). By contrast, the module with the second most significant sex-association, the Magenta module, which contains 544 genes, showed male-bias, as well as decreases with aging (**Fig 2B**). To unbiasedly determine which functions were associated with the Salmon and Magenta module, we used hypergeometric enrichment with Gene Ontology and Reactome categories (**Fig 2C and 2D**; **S3 Table**). Consistent with the presence of collagen-related genes, the Salmon module was strongly associated with collagen- and ECM-related terms (**Fig 2C and 2D**; **S3 Table**). By contrast, the Magenta module was strongly associated with chromatin and cell cycle-related terms (**S3 Table**), consistent with our previous

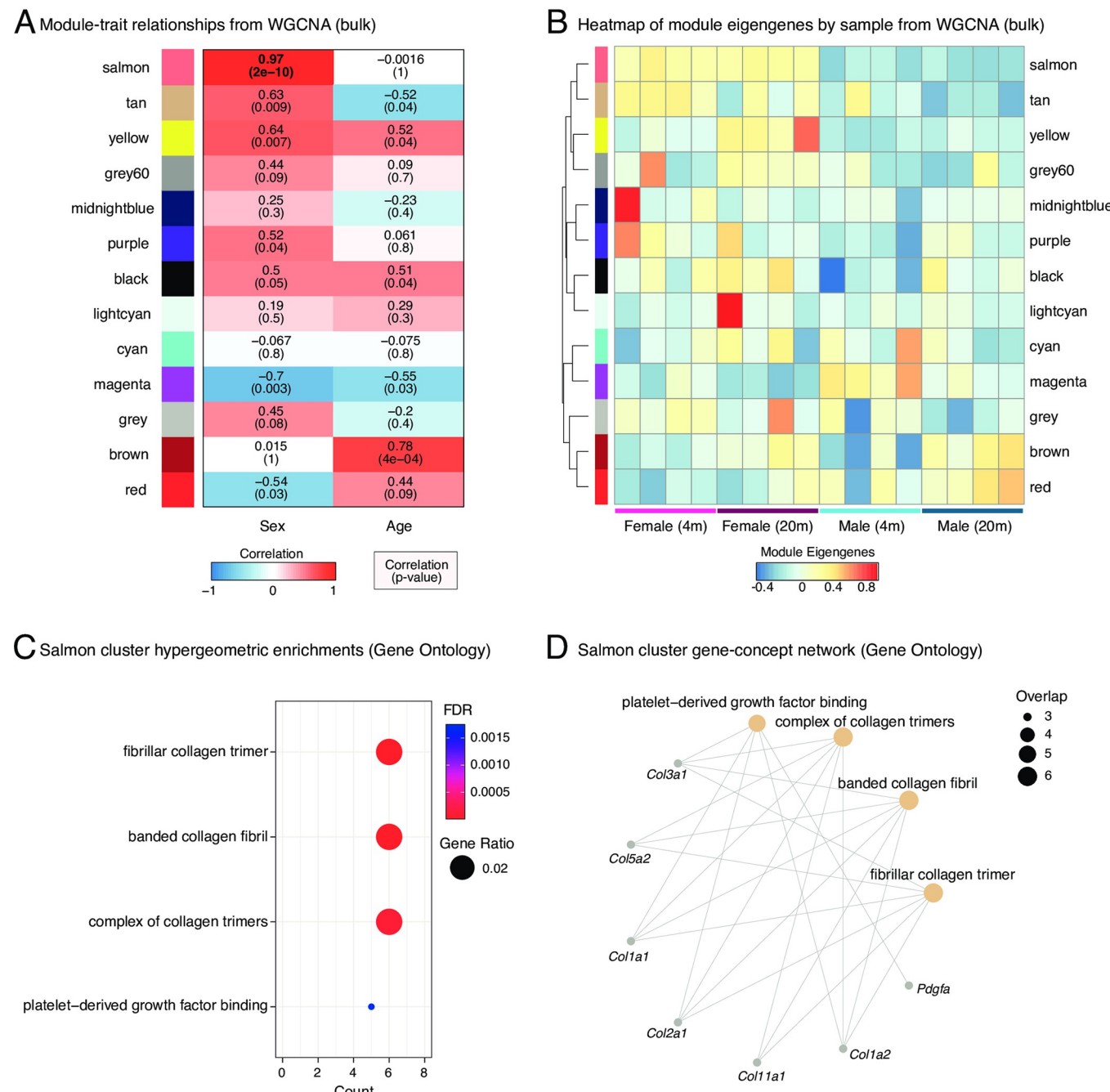

**Fig 2. WGCNA reveals that the top female-biased expression module is enriched for collagen and ECM-related genes.** (A) Module-trait relationships from neutrophil bulk RNA-seq WGCNA. Note that the Salmon module (bold) is the most biased for female neutrophils. Also see **S1 Fig** and **S3 Table.** (B) Heatmap of sample-wise module eigengenes from bulk neutrophil RNA-seq WGCNA analysis. (C) Dotplot of significantly enriched gene sets associated to the Salmon module, using hypergeometric enrichment analysis with Gene Ontology [GO] terms (FDR < 0.05). Also see full results in **S3 Table**. (D) Gene Concept network for GO enrichment analysis related to the Salmon cluster, derived from the same analysis as (C).

observations that these terms were male-biased and regulated with aging [11]. Together, our analysis of the topmost sex-associated WGCNA modules revealed that ECM gene expression is a sex-dimorphic transcriptional feature of neutrophils.

## Discussion

While neutrophils, upon activation, are known to secrete elastase and MMPs to remodel ECM [13, 15], activate fibroblasts [24, 25], or even transport ECM to wound sites [16], little is known about the intrinsic expression of ECM genes of neutrophils in their bone marrow niche. Here, we show that the most significant gene cluster correlating with biological sex (Salmon cluster) is enriched with a distinct set of ECM genes that are biased to be expressed at higher levels in female neutrophils. With our analysis, we were not able to detect an impact of aging on the regulation of ECM-related gene expression (regardless of sex). Nevertheless, it is possible that changes related to aging as a function of sex may emerge in datasets with larger numbers of animals, which should be investigated in future studies. In addition, future single cell RNA-seq studies of neutrophils including female and male mice across the lifespan may provide important new insights into the impact of aging and sex on neutrophil heterogeneity, including relating to ECM biology.

We identified potential gene sets in the Salmon cluster that might prime female neutrophils for improved wound healing (**S3 Table**). This gene network consists of collagens interacting with platelet-derived growth factor binding (**Fig 2**), known to initiate neutrophil activation and inflammatory responses at sites of injury [26]. Interestingly, integrin a5 (Itga5) was uniquely expressed in the Salmon cluster. Neutrophils that express α5β1 integrin bind fibronectin [27]. Fibronectin is important for wound healing and is required to be placed first in order to add collagen to build a new functional ECM [28]. In human bone fracture hematomas, neutrophils localize to fibronectin shortly after injury, and hematoma-localized neutrophils are uniquely stained for cellular fibronectin [17]. This suggests that activated neutrophils in the hematoma may produce a first "emergency ECM" [17], consistent with our observation of fibrillar collagen gene expression of neutrophils. Furthermore, upon adhering to fibronectin, neutrophils secrete branched-chain (valine, isoleucine, leucine), aromatic (tyrosine, phenylalanine), and positively charged free amino acids (arginine, ornithine, lysine, hydroxylysine, histidine) [17]. Although the reason for this is unclear, the release of, for instance, hydroxylysine, crucial for Lysol oxidase-mediated collagen crosslinking, is either increased by stimulating neutrophils with insulin or reduced by blocking the PI3K/Akt pathway [17]. Based on this and the composition of the female-enriched Salmon module, we speculate on a connection among this metabolic axis via serum/glucocorticoid regulated kinase 1 (Sgk1), integrin a5, and fibrillar collagens (Col1a1/2, Col2a1), integrated stress response activating transcription factor 4 (Atf4), also implicated in amino acid metabolism for collagen biosynthesis [29], and potentially modulating immune responses via Pdgfa and interleukin 6 receptor alpha (Il6ra) and further collagen synthesis fibroblast growth factor receptor 1 (Fgfr1) (**S3 Table**). This hypothesis might be interesting to test in the future and might provide a mechanistic explanation for why female mice show enhanced wound-healing compared to males [30, 31].

Understanding neutrophil biology has broad implications for therapeutic interventions. Additionally, investigating sex-dimorphic mechanisms affecting neutrophil responses is vital for developing interventions beneficial to both males and females. For example, there is a clear female bias when it comes to rheumatoid arthritis, whereby 75% of patients are women [32]. Inducing arthritis in mice by anti-type II collagen antibodies and lipopolysaccharide, leads to massive neutrophil infiltration into the joint space [33]. Fascinatingly, depleting neutrophils either by the onset of arthritis or during arthritis completely ameliorates the disease [33], suggesting that neutrophils not only initiate but are also involved in maintaining arthritis disease progression. Thus, investigating female-biased processes in neutrophils, including the contribution of the ECM to promote pathology, might be an important next step in managing

rheumatoid arthritis. These female-biased signatures suggest that targeting sex hormone signaling could be a potential therapeutic translation of these findings.

This analysis offers new insights into the potential molecular underpinning of sex-differences in ECM biology. Our analysis, which focuses on differential expression of ECM-related gene sets, reveals a potential source for female-biased ECM production, which has not been previously explored and provides new insights for the scientific community to pursue. While our research primarily hinges on transcriptomic data, it does not offer direct insights into protein levels or functional outcomes. Future studies using targeted CRISPR or shRNA screens in neutrophil cell lines (e.g. MPRO) or use of transgenic mouse lines carrying mutations in ECM-related genes will be useful to understand the role of the clusters found in our data in setting sex-dimorphic physiological responses of neutrophils. Additionally, the intricate regulation of post-transcriptional and post-translational mechanisms may contribute substantial influence over the ECM, thus necessitating further investigation. Our study focuses on neutrophils, which are one part of the complex immune system, and our analyses do not encompass their interactions with other cells, thereby limiting the overall understanding of these intricate relationships. Ultimately, future studies investigating the mechanisms by which the ECM influences neutrophil activity and its interaction with other immune cells are essential, as well as examining how these features are conserved or differ from human neutrophils.

In summary, we examined the sex-specific gene expression differences of neutrophils. We established that a major difference between male and female neutrophils is the expression of ECM genes. We identified a gene network that might prime ECM gene expression in female neutrophils to potentially accelerate wound healing. Dissecting sex-specific and intrinsic gene expression of neutrophils may significantly impact our understanding to tailor sex-specific therapeutic interventions for wound healing, immune responses, and arthritis.

## Methods

### Data acquisition and preprocessing

We obtained published processed bulk RNA-seq differential expression results from young (4 months) and aged (20 months), male and female primary bone marrow neutrophils derived from C57BL/6JNia mice (PRJNA630663; **Fig 1A** upper; n = 4/group) [11]. For all analyses below, we used the SVA-normalized count table and DEseq2 analyses results generated in that study, for which the processed files were available online from https://github.com/BenayounLaboratory/Neutrophil_Omics_2020 [11].

We also obtained a processed and annotated Seurat object containing data from a 10xGenomics single-cell RNA-seq dataset of young (3 months) male and female primary bone marrow neutrophils derived from C57BL/6J mice (PRJNA796634; **Fig 1A** lower; n = 2/group) [12]. This file was obtained from Figshare (https://doi.org/10.6084/m9.figshare.19623978) [12]. R package 'Seurat' 4.2.0 was used to load and interact with the annotated single-cell RNA-seq expression dataset.

Finally, we obtained a processed serum proteomics dataset from C57BL/6NHsd female and male mice (4–5 months of age) [22]. The processed proteomics expression file was obtained from S2 Table from the publisher's site (see below data availability statement). For our analysis, we only considered label-free quantitation [LFQ] intensity data derived from wild-type animals. We performed Multidimensional Scaling (MDS) analysis on protein expression levels in serum using a distance metric between samples based on the Spearman's rank correlation value (1-Rho), then provided to the core R command 'cmdscale'. Limma 3.50.3 [23] was used to perform differential expression analysis, with sex as the variable of interest and ascorbate treatment as a modeling covariate. Proteins with Limma False Discovery Rate [FDR] < 0.05 were considered significantly sex-biased and are reported in **S2D Table**.

### Curated lists of extracellular matrix-related terms

We obtained a manually curated list of gene sets relevant to extracellular matrix biology [18]. The curated gene sets are also provided in **S1 Table.**

**Gene Set Enrichment Analysis of extracellular matrix-related terms.** We used the GSEA paradigm to determine whether ECM-related gene sets were differentially regulated as a function of sex or age in bulk RNA-seq of primary neutrophils [34]. For this purpose, we used R package 'phenotest' 1.42.0 in R version 4.1.2. The DEseq2 t-statistic for each gene calculated for age- or sex-expression was used to create the ranked gene list for functional enrichment analysis for both sex and aging [11]. The table output of this analysis is reported in **S2A and S2B Table** and graphically represented in **Fig 1B**.

### Overrepresentation analysis using Fisher's exact test

To determine the significance of the overlap between significantly female-biased genes in neutrophils by RNA-seq (DEseq2 FDR < 0.05) or serum proteomics (limma FDR < 0.05) and ECM-related gene lists, we used Fisher's exact test. The background used was all genes detected by RNA-seq or proteins detected by proteomics, respectively. P-value correction for multiple hypothesis testing was performed using the Benjamini-Hochberg method. The table output of this analysis is reported in **S2C and S2E Table** and graphically represented in **S1A and S1E Fig**.

To determine the significance of the overlap between X-linked genes and genes from the Salmon module, we obtained a list of X-encoded genes from ENSEMBL BioMart version 108 (database access 2023/02/02), restricting output based on gene position on chromosome X. This list was further restricted to genes expressed in our bulk neutrophil RNA-seq according to our previous study [11]. The Fisher's exact test was used to determine the significance of the overlap between genes of the Salmon module and expressed X-linked genes, using the list of expressed genes from the bulk RNA-seq as the universe.

### Ucell scoring analysis of extracellular matrix-related terms

To determine whether ECM-related terms were differentially regulated as a function of neutrophil biological sex at the single-cell level, we leveraged the UCell robust single-cell gene signature scoring metric implemented through R package 'UCell' 1.3.1 [20]. Cell-wise UCell scores were computed for each ECM-related gene list and are graphically reported as aggregates by sample of origin in **Fig 1C**. Note that Collagen-related genes were not robustly detected at the single cell level, likely a limitation of the platform, and have an extremely low percentage of cells with detected expression (**Fig 1C**). For analysis of statistical significance, a non-parametric Wilcoxon rank sum test was used to compare the distribution of UCell scores between female and male neutrophils and is reported for each gene set in **Fig 1C**.

### Weighted gene co-expression network analysis [WGCNA]

To investigate what other genes are being co-expressed with the collagens and other ECM-related genes, we used the WGCNA paradigm [35, 36]. We used R package 'WGCNA' 1.71 [36] and the above-mentioned normalized SVA count table as the inputs. The data was preprocessed by removing any genes with inconsistent expression, as recommended by the user manual. A power of 10 was chosen to create the scale-free topology based on the power and scale independence equilibration. The produced topological overlap matrix [TOM] reveals the mean network connectivity of each gene, and genes with similar expression profiles were then classified into different modules. We set the minimum module size to 100. The analyses yield

13 significant modules. We then performed module-trait analysis using age and sex as the traits to correlate to the modules.

## Functional enrichment analysis of WGCNA gene modules

We performed functional enrichment analysis for genes associated with the top sex-biased module, the 'Salmon' module (p = 2x10^-10; **Fig 2A and 2B**), to unbiasedly determine the top functions differentially regulated in primary neutrophils as a function of sex. We used a hyper-geometric test to determine overrepresented Gene Ontology [GO] terms and REACTOME Genesets in the Salmon module compared to expressed genes in the bulk neutrophil transcriptome dataset. For this purpose, we took advantage of R package 'ClusterProfiler' 4.2.2 [37], with GO annotations derived from Bioconductor annotation package 'org.Mm.eg.db' 3.14.0, and with Reactome annotations derived from the 'Reactome-PA' 1.38.0 package [38]. Both dotplot and gene concept network results were derived from ClusterProfiler plotting functions.

## Supporting information

**S1 Fig. Analysis of ECM-related gene expression in bulk neutrophil RNA-seq, single-cell neutrophil RNA-seq and serum proteomics.** (A) Bubble plot showing Fisher's exact test enrichment results for overlap of significantly female-biased genes in neutrophil bulk transcriptomes (DESeq2 FDR < 5%) and ECM-related genes. See full results in **S2C Table**. (B) Dotplot of 'Ucell' scores of ECM-related genesets aggregated by neutrophil maturation subset (as defined by [21]) in single-cell neutrophil RNA-seq dataset from [12]. (C) Multidimensional scaling analysis for mouse serum proteomics. MDS: Multidimensional Scaling. (D) Heatmap of significant (Limma FDR < 5%) sex-dimorphic proteins in mouse serum. Also see **S2D Table**. (E) Bubble plot showing Fisher's exact test enrichment results for overlap of significantly female-biased proteins in serum proteomics (Limma FDR < 5%) and ECM-related proteins. See full results in **S2E Table**.
(TIF)

**S2 Fig. WGCNA module cluster dendrogram and salmon module expression profile.** (A) WGCNA module cluster dendrogram from bulk neutrophil RNA-seq. (B) Heatmap of WGCNA Salmon module gene expression from bulk neutrophil RNA-seq. **S1 Table**: Curated Gene Lists for ECM-related genes.
(TIF)

**S1 Table. Curated Gene Lists for ECM-related genes.**
(XLSX)

**S2 Table. Enrichment analysis results for ECM-related gene lists.** (A) GSEA results for ECM-related gene lists on bulk neutrophil RNA-seq, as a function of age (sex as covariate). Note that no ECM gene set passed significance thresholds for age-related expression. (B) GSEA results for ECM-related gene lists on bulk neutrophil RNA-seq in female vs. male neutrophils (age as covariate). (C) Analysis of significance of overlap for female-biased neutrophil genes by RNA-seq (DEseq2 FDR < 0.05) for ECM related gene list. (D) Limma differential expression analysis of sex-biased proteins in C57BL/6 serum proteomics from [22] (FDR < 0.05). (E) Analysis of significance of overlap for female-biased serum proteins by proteomics (Limma FDR < 0.05) for ECM-related gene list.
(XLSX)

**S3 Table. WGCNA related analysis with module membership, and Gene Ontology and Reactome hypergeometric enrichment analysis of Salmon and magenta modules.**
(XLSX)

## Acknowledgments

Some panels created with BioRender.com.

## Author Contributions

**Conceptualization:** Collin Y. Ewald, Bérénice A. Benayoun.

**Data curation:** Bérénice A. Benayoun.

**Formal analysis:** Cassandra J. McGill, Bérénice A. Benayoun.

**Funding acquisition:** Collin Y. Ewald, Bérénice A. Benayoun.

**Investigation:** Cassandra J. McGill, Bérénice A. Benayoun.

**Supervision:** Bérénice A. Benayoun.

**Writing – original draft:** Cassandra J. McGill, Collin Y. Ewald, Bérénice A. Benayoun.

**Writing – review & editing:** Cassandra J. McGill, Collin Y. Ewald, Bérénice A. Benayoun.

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
