## [Decision Letter · Decision Letter 0]

6 Sep 2023

PONE-D-23-16855Sex-dimorphic expression of extracellular matrix genes in mouse bone marrow neutrophilsPLOS ONE

Dear Dr. Benayoun,

Thank you for submitting your manuscript to PLOS ONE. After careful consideration, we feel that it has merit but does not fully meet PLOS ONE’s publication criteria as it currently stands. Therefore, we invite you to submit a revised version of the manuscript that addresses the points raised during the review process.

Your manuscript offers a valuable examination of sexual dimorphism in the innate immune system, with an emphasis on neutrophils and the ECM component. The potential therapeutic implications based on sex-specificities highlighted are noteworthy.

However, several points warrant revision:

1. Methods: Clarify the number of animals used in prior studies that provided the RNA-seq data.

2. Data Expansion: Include single-cell RNA-seq data from older mice to enhance age-based insights.

3. Data Justification: Justify the choice of serum proteomics data, given its divergence from the primary RNA-seq data source.

We await your revised submission for another review round.

We look forward to receiving your revised manuscript.

Kind regards,

Syed M. Faisal, Ph.D.

Academic Editor

PLOS ONE

“Some panels created with BioRender.com. This work was supported by NIA T32 AG052374 predoctoral fellowship to C.J.M., by the Swiss National Science Foundation Funding from the SNF P3 Project 190072 to CYE, NIA R01 AG076433 to B.A.B. and Pew Biomedical Scholar award #00034120 to B.A.B.”

“This work was supported by National Institute of Aging [https://www.nia.nih.gov/] T32

AG052374 predoctoral fellowship to C.J.M., by the Swiss National Science Foundation

[https://www.snf.ch/en] from the SNF P3 Project 190072 to CYE, National Institute of

Aging [https://www.nia.nih.gov/] R01 AG076433 to B.A.B. and Pew Biomedical Scholar

award #00034120 from the Pew Charitable Trust [https://www.pewtrusts.org/] to B.A.B.

The funders had no role in study design, data collection and analysis, decision to

publish, or preparation of the manuscript.”

4. We note that Figure 1A in your submission contain copyrighted images. All PLOS content is published under the Creative Commons Attribution License (CC BY 4.0), which means that the manuscript, images, and Supporting Information files will be freely available online, and any third party is permitted to access, download, copy, distribute, and use these materials in any way, even commercially, with proper attribution. For more information, see our copyright guidelines: http://journals.plos.org/plosone/s/licenses-and-copyright.

1. You may seek permission from the original copyright holder of Figure 1A to publish the content specifically under the CC BY 4.0 license.

Reviewers' comments:

Reviewer's Responses to Questions

**Comments to the Author**

1. Is the manuscript technically sound, and do the data support the conclusions?

Reviewer #1: Partly

Reviewer #2: Yes

Reviewer #3: Yes

Reviewer #4: Yes

2. Has the statistical analysis been performed appropriately and rigorously? 

Reviewer #1: Yes

Reviewer #2: Yes

Reviewer #3: Yes

Reviewer #4: Yes

3. Have the authors made all data underlying the findings in their manuscript fully available?

Reviewer #1: Yes

Reviewer #2: Yes

Reviewer #3: Yes

Reviewer #4: Yes

4. Is the manuscript presented in an intelligible fashion and written in standard English?

Reviewer #1: Yes

Reviewer #2: Yes

Reviewer #3: Yes

Reviewer #4: Yes

5. Review Comments to the Author

Reviewer #1: In their manuscript entitled: “Sex-dimorphic expression of extracellular matrix genes in mouse bone marrow neutrophils” McGill and colleagues describe the follow up analysis stemming from their previous work, on the sex-dimorphic expression, in neutrophils, of extra-cellular matrix proteins and of the associated enzymes and other proteins.

The manuscript is generally well written, with all results available as supplementary tables, and the analyses presented mostly support the conclusions reached by the authors. However, it is unclear how the paucity of expression of the different collagen genes in single cell does affect the analysis. It is also a pity to see that there is no analysis of publicly available human datasets (Chen et al, DOI: 10.1016/j.cell.2016.10.026 or https://maayanlab.cloud/archs4/download.html ) to confirm that the findings made using mouse as a model system can be extended to humans and therefore being relevant for human health, as the authors conclude in their discussion.

Few minor criticisms:

1) neutrophils are the most abundant white blood cell type but not the most abundant blood cell type.

2) I would suggest avoiding the use of adjectives that are open to subjective interpretation, such as tantalising.

3) Instead of (i.e. matrisome), I would use (hereafter matrisome)

4) Figure 1, probably unintentional but I would suggest stopping the perpetuating the stereotype to assign the colours pink and blue to discriminate between females and males.

5) Lines 102/122 Fisher’s exact test is used but FDR results are shown, could the authors please explain which is correct?

6) WGCNA analysis, I am well aware that the colours are the output of the algorithm. However, to support colour blind colleagues and other readers it would be better to replace colours with numbers.

7) Figure 2C, the gene ration legend is incomplete.

Reviewer #2: The research article “Sex-dimorphic expression of extracellular matrix genes in mouse bone marrow neutrophils” by the authors is a well thought analysis that highlights the importance of sexual dimorphism in the innate immune system. The authors have exploited the already available RNA-seq data (bulk and single-cell) from previous studies showing sexual dimorphism in one of the most abundant leukocytes i.e., neutrophils. They used these data to further analyze the ECM component and how they are expressed in a sexually biased way.

The reason for focusing on the ECM transcriptomic profile is that ECM is an important part of innate immune regulation, and it was among the top sex-dimorphic genes along with the collagen synthesis genes as observed from the previous bulk RNA-seq data analyses.

They confirmed that there is female bias in the expression of genes related to ECM and its components in the neutrophils from both young and old animals. The data from single-cell RNA-seq also gave female-biased enrichment for the ECM. Further, the WGCNA analysis also was observed to be female-biased with significant enrichment of ECM-related genes.

The study holds significance in terms of sex-specific immune regulation and the potential for translating into therapeutic interventions based on the sex of the individuals.

The article is well-written and explained. Some questions need to be addressed.

1. In the methods section, authors should add the number of animals used in the previous studies from where the RNA-seq data was obtained.

2. Why the single-cell RNA-seq data was generated only for the young mice and not the older mice? The authors are talking about the sex-dimorphic heterogeneity throughout the age of the mice. They should also get the single-cell RNA-seq data from the older animals and do the analysis. This will add value to the data and might substantiate the present results or might give some interesting findings.

3. Authors have used serum proteomics data as the mouse neutrophil quantitative proteomics data was not available. How do authors justify the use of serum proteomics data? The neutrophil proteins in the serum are from circulating neutrophils and will only be the secretory proteins while the RNA-seq data are from the primary bone marrow neutrophils (naive).

Reviewer #3: In this article, authors investigated the sex-specific disparity in gene expression of neutrophils.

This study reports a remarkable difference in the expression of extracellular matrix genes between male

and female neutrophils and established that gene expression of extracellular matrix is a sex-

dimorphic transcriptional feature of neutrophils and the expression of genes is female biased.

The authors also recognized significant sex-biased gene expression modules in neutrophils. The

manuscript is well-written and the experiments are skillfully designed. I have only a few minor concerns

about the manuscript.

Abbreviations should be defined where used for the first time and then be consistent throughout the

manuscript.

Remove second “the” from line 133, Page 6.

Add Xie et al. (2020) before [21]. Line 137, Page 6.

Reviewer #4: The manuscript by Cassandra J. McGill et al., titled " Sex-dimorphic expression of extracellular matrix genes in mouse bone marrow neutrophils," has been well written and study aimed to investigate the effects of innate immune system in mammals displays sex-based differences, with neutrophils, a key type of white blood cell, constituting the front-line defense. The primary focus of this study is the extracellular matrix (ECM)-related genes emerged as prominent contributors among these sex-specific genes. The author has delved into the transcriptomic patterns of primary mouse bone marrow neutrophils on both bulk and single-cell levels and found that ECM's has role in governing innate immune responses. Furthermore, this study found female-biased expression in mouse neutrophil ECM gene clusters using existing ECM gene sets and weighted gene co-expression network analysis. In summary, author found the increased prevalence of immune-related disorders like rheumatoid arthritis in women, these findings may provide light on sex-influenced inflammatory diseases.

While the study provides valuable insights into the differential expression of extracellular matrix (ECM)-related genes in neutrophils based on organismal age and biological sex, there are several limitations to consider:

Limitation:

• The study focuses primarily on transcriptomic data. While gene expression levels provide important information, they do not directly reflect protein levels or functional outcomes. Post-transcriptional and post-translational regulatory mechanisms could influence the actual ECM-related protein production and function.

• Justify the fact that in this study's author only focus on neutrophils and ECM-related genes represents a simplified view of the complex interactions occurring within the immune system and the extracellular matrix. Neutrophils are just one component of a broader cellular network, and their interactions with other immune cells are not considered under this study.

• The study focuses predominantly on gene expression and lacking functional validation of the roles of identified ECM-related genes in neutrophil biology, ECM remodeling, and disease pathogenesis.

• The study provides a descriptive analysis of gene expression patterns but lacks mechanistic insights into how these genes contribute to neutrophil function, ECM remodeling, and disease outcomes, discuss this as limitation of this study?

Minor comments:

• Explain why this study hints at potential implications for managing diseases such as rheumatoid arthritis by investigating female-biased processes in neutrophils, including ECM-related factors. How can these findings be translated to clinical applications for improving therapeutic interventions in conditions involving neutrophil-mediated responses?

• Discussion underscores the importance of understanding sex-specific gene expression differences in neutrophils to develop tailored therapeutic strategies. How can the insights gained from this study be leveraged to design interventions that consider both male and female-specific responses?

• While the study identifies ECM-related gene clusters, functional validation is lacking. How can these genes' roles be experimentally confirmed in modulating neutrophil behavior, ECM remodeling, and disease outcomes, Need explanation?

• The study discusses the interplay between neutrophil activation, ECM deposition, and wound healing. How does neutrophil activation status affect ECM-related gene expression, and how does this interaction influence various disease contexts? Cite some study.

• Write the brief paragraph regarding the limitation of this study.

6. PLOS authors have the option to publish the peer review history of their article (what does this mean?). If published, this will include your full peer review and any attached files.

Reviewer #1: No

Reviewer #2: No

Reviewer #3: **Yes: **Nazim Husain

Reviewer #4: **Yes: **ZEESHAN AHMAD

---

## [Author Response · Author response to Decision Letter 0]

12 Oct 2023

Editorial comments:

1. Methods: Clarify the number of animals used in prior studies that provided the RNA-seq data.

We have updated the methods section to include the number of animals from the bulk RNA-seq and single cell RNA-seq datasets: “We obtained published processed bulk RNA-seq differential expression results from young (4 months) and aged (20 months), male and female primary bone marrow neutrophils derived from C57BL/6JNia mice (PRJNA630663; Fig 1A upper; n = 4/group)” (page 12, line 264-266), “We also obtained a processed and annotated Seurat object containing data from a 10xGenomics single-cell RNA-seq dataset of young (3 months) male and female primary bone marrow neutrophils derived from C57BL/6J mice (PRJNA796634; Fig 1A lower; n = 2/group)” (page 12, line 270-272).

2. Data Expansion: Include single-cell RNA-seq data from older mice to enhance age-based insights.

Although a new dataset including young and old animals would be very interesting, we believe it is outside of the scope of this study, a point which we raised with the editor Dr. Faisal by email communication. 

Indeed, our current study is strictly a reanalysis of publicly available datasets. As the points we make in this study are not relevant to aging and only to sex-dimorphism, generating this new dataset would not change our conclusions regarding to sex-dimorphism. 

However, to recognize the future value of such a dataset, we now include a new sentence in the discussion: “In addition, future single cell RNA-seq studies of neutrophils including female and male mice across the lifespan may provide important new insights into the impact of aging and sex on neutrophil heterogeneity, including relating to ECM biology.” (page 9, line 197-200).

3. Data Justification: Justify the choice of serum proteomics data, given its divergence from the primary RNA-seq data source.

Since neutrophils represent a large portion of circulating leukocytes and are thus likely to contribute significantly to serum protein production, we reasoned that neutrophil-driven differences should be detectable in the serum proteome. Although the proteomics data may reflect secretion from circulating neutrophils rather than bone marrow neutrophils, we believe the difference is likely to be minute. Indeed, neutrophils complete differentiation in the bone marrow, enter circulation, and home back to the bone marrow. Specifically, mature neutrophils migrate back to the bone marrow and a substantial portion of bone marrow neutrophils are not naïve [PMID: 32719519]. That results in the mature neutrophil fraction (which is the one predominant in circulation) being represented at substantial levels in the bone marrow fraction (as much as ~30%).

We have now adjusted the text to say, “While neutrophils are likely contributing to this sex-bias in serum ECM proteomics signatures, this observation may also indicate a more general sex bias in ECM expression among immune/circulating cells” (page 7, line 153-155). 

 

Reviewer’s comments: 

Reviewer #1: 

In their manuscript entitled: “Sex-dimorphic expression of extracellular matrix genes in mouse bone marrow neutrophils” McGill and colleagues describe the follow up analysis stemming from their previous work, on the sex-dimorphic expression, in neutrophils, of extra-cellular matrix proteins and of the associated enzymes and other proteins.

The manuscript is generally well written, with all results available as supplementary tables, and the analyses presented mostly support the conclusions reached by the authors. 

We thank the reviewer for recognizing that our analyses support our conclusions.

1. However, it is unclear how the paucity of expression of the different collagen genes in single cell does affect the analysis. 

It is a known problem that some genes that are expressed drop out with single cell RNA-seq; thus, the number of genes we detect are lower and we are unable to run enrichment [PMID: 32718323, PMID: 32762710]. We updated the text to explicitly indicate this issue “which makes single-cell level analysis not robust for this class of genes” (page 6, line 142-143).

2. It is also a pity to see that there is no analysis of publicly available human datasets (Chen et al, DOI: 10.1016/j.cell.2016.10.026 or https://maayanlab.cloud/archs4/download.html ) to confirm that the findings made using mouse as a model system can be extended to humans and therefore being relevant for human health, as the authors conclude in their discussion.

Immune cells are known to vary quite a lot between species [PMID: 14978070], which unfortunately makes direct comparisons not very meaningful, so we made a deliberate choice to focus analyses in this paper to mouse bone marrow neutrophils. Thus, we believe that analysis of human data (for which directly comparable datasets are not available) is outside of the scope of the current manuscript. 

While we agree that a similar study in human cells may hold interest, we believe that focusing our analysis on mouse data is more rigorous, and we disclose explicitly in the title of our manuscript that this analysis was done exclusively on mouse datasets. We also make sure that this is explicitly stated in the abstract of our manuscript.

To address this in the manuscript, we added this sentence in the discussion “Ultimately, future studies investigating the mechanisms by which the ECM influences neutrophil activity and its interaction with other immune cells are essential, as well as examining how these features are conserved or differ from human neutrophils” (page 10, line 249-252).

3. Neutrophils are the most abundant white blood cell type but not the most abundant blood cell type.

We adjusted the text to now say “Neutrophils are the most abundant white blood cell type in human blood, representing 50-70% of leukocytes in humans throughout life” (page 3, line 45).

4. I would suggest avoiding the use of adjectives that are open to subjective interpretation, such as tantalising.

We adjusted the text to now say “To follow-up on this observation of sex-dimorphism in ECM gene expression, we reanalyze transcriptomic data from primary mouse bone marrow neutrophils to help understand how biological sex may influence ECM component gene expression.” (page 4, line 74).

5. Instead of (i.e. matrisome), I would use (hereafter matrisome)

We changed (i.e. matrisome) to (hereafter matrisome) (page 5, line 104). 

6. Figure 1, probably unintentional but I would suggest stopping the perpetuating the stereotype to assign the colours pink and blue to discriminate between females and males.

While we understand that this color choice might feel gender normative, we believe that for the expediency of scientific communication it is important to choose a color palette that is easily understandable without confusion. We also use the same color scheme across all publications from the lab for consistency, which is crucial for scientific rigor and reproducibility.

Finally, we also believe this is a stylistic rather than scientific choice, that belongs to us as authors. We apologize if this is offensive to the reviewers.

Nevertheless, we choose to keep the pink/blue scheme for consistency of our body of work. 

7. Lines 102/122 Fisher’s exact test is used but FDR results are shown, could the authors please explain which is correct?

We apologize for the lack of clarity in our initial phrasing. 

In this case, we use the Fisher exact test to evaluate the significance of overlap between ECM gene sets and significantly regulated genes from DESeq2. FDR in that context refers to genes that passed significance for differential gene expression in DESeq2 (not related to Fisher). To make this clearer in the revised manuscript, we have updated the text to now say “Genes biased for female expression according to DESeq2 at FDR < 5%” (page 6, line 123-124).

8. WGCNA analysis, I am well aware that the colours are the output of the algorithm. However, to support colour blind colleagues and other readers it would be better to replace colours with numbers.

We respectfully disagree with the reviewer that our WGCNA figures are not compatible with color blind individuals. Indeed, all modules are always labelled with their color AND color name in Fig 2, which means that color vision does not affect figure readability. This is also why we provide the complete list with color name of the module and genes in the module as a supplementary table.

For Fig S2, while color blindness may further complicate reading of the panel, it is a difficult type of figure to grasp even for color-able people, and was only provided as a supplement. We do not believe this impacts the ability of color-blind scientists to understand the methods, results, and conclusions of our study. 

9. Figure 2C, the gene ration legend is incomplete.

While the reviewer intuits that part of the legend is missing, this is not the case. The panel was reformatted in figure making for adherence to figure guidelines, but the legend was not cut. 

Indeed, we provide a copy of the raw output from clusterprofiler below. As the reviewer can see, the legend in Figure 2C is reproduced faithfully from this output. We hope this clarifies the confusion.

 

Reviewer #2: 

The research article “Sex-dimorphic expression of extracellular matrix genes in mouse bone marrow neutrophils” by the authors is a well thought analysis that highlights the importance of sexual dimorphism in the innate immune system. The authors have exploited the already available RNA-seq data (bulk and single-cell) from previous studies showing sexual dimorphism in one of the most abundant leukocytes i.e., neutrophils. They used these data to further analyze the ECM component and how they are expressed in a sexually biased way.

The reason for focusing on the ECM transcriptomic profile is that ECM is an important part of innate immune regulation, and it was among the top sex-dimorphic genes along with the collagen synthesis genes as observed from the previous bulk RNA-seq data analyses.

They confirmed that there is female bias in the expression of genes related to ECM and its components in the neutrophils from both young and old animals. The data from single-cell RNA-seq also gave female-biased enrichment for the ECM. Further, the WGCNA analysis also was observed to be female-biased with significant enrichment of ECM-related genes.

The study holds significance in terms of sex-specific immune regulation and the potential for translating into therapeutic interventions based on the sex of the individuals.

The article is well-written and explained. Some questions need to be addressed.

We thank the reviewer for their kind words about our manuscript and for noting the importance of our findings for the broader community. 

1. In the methods section, authors should add the number of animals used in the previous studies from where the RNA-seq data was obtained.

We have now added the number of animals to the methods section (page 12, lines 266/272).

2. Why the single-cell RNA-seq data was generated only for the young mice and not the older mice? The authors are talking about the sex-dimorphic heterogeneity throughout the age of the mice. They should also get the single-cell RNA-seq data from the older animals and do the analysis. This will add value to the data and might substantiate the present results or might give some interesting findings.

Although a new dataset including young and old animals would be very interesting, we believe it is outside of the scope of this study, a point which we raised with the editor Dr. Faisal by email communication. 

Indeed, our current study is strictly a reanalysis of publicly available datasets. As the points we make in this study are not relevant to aging and only to sex-dimorphism, generating this new dataset would not change our conclusions regarding to sex-dimorphism. 

However, to recognize the future value of such a dataset (as suggested by the reviewer), we now include a new sentence in the discussion: “In addition, future single cell RNA-seq studies of neutrophils including female and male mice across the lifespan may provide important new insights into the impact of aging and sex on neutrophil heterogeneity, including relating to ECM biology.” (page 9, line 197-200).

3. Authors have used serum proteomics data as the mouse neutrophil quantitative proteomics data was not available. How do authors justify the use of serum proteomics data? The neutrophil proteins in the serum are from circulating neutrophils and will only be the secretory proteins while the RNA-seq data are from the primary bone marrow neutrophils (naive).

We apologize for not making the reasons of our choice clearer.

Since neutrophils represent a large portion of circulating leukocytes and are thus likely to contribute significantly to serum protein production, we reasoned that neutrophil-driven differences should be detectable in the serum proteome. Although the proteomics data may reflect secretion from circulating neutrophils rather than bone marrow neutrophils, we believe the difference is likely to be minute. Indeed, neutrophils complete differentiation in the bone marrow, enter circulation, and home back to the bone marrow. Specifically, mature neutrophils migrate back to the bone marrow and a substantial portion of bone marrow neutrophils are not naïve [PMID: 32719519]. That results in the mature neutrophil fraction (which is the one predominant in circulation) being represented at substantial levels in the bone marrow fraction (as much as ~30%).

We have now adjusted the text to say, “While neutrophils are likely contributing to this sex-bias in serum ECM proteomics signatures, this observation may also indicate a more general sex bias in ECM expression among immune/circulating cells” (page 7, line 153-155). 

 

Reviewer #3: 

In this article, authors investigated the sex-specific disparity in gene expression of neutrophils.

This study reports a remarkable difference in the expression of extracellular matrix genes between male and female neutrophils and established that gene expression of extracellular matrix is a sex-

dimorphic transcriptional feature of neutrophils and the expression of genes is female biased.

The authors also recognized significant sex-biased gene expression modules in neutrophils. The

manuscript is well-written and the experiments are skillfully designed. I have only a few minor concerns about the manuscript.

We thank the reviewer for their positive comments on the rigor of our manuscript. We thank them for their suggestions and have incorporated all of them into our improved manuscript. 

1. Abbreviations should be defined where used for the first time and then be consistent throughout the manuscript.

We fixed instances of defining abbreviations twice (page 5, line 96; page 7, line 165), and ensured this was consistent throughout the manuscript. 

2. Remove second “the” from line 133, Page 6.

We removed the extra “the” in the sentence “We show the summarized UCell scoring values for each ECM-related gene set per biological sample, and the significance of the difference in score distributions in female vs. male neutrophils” (page 6, line 134-136). 

3. Add Xie et al. (2020) before [21]. Line 137, Page 6.

We added “Xie et al. (2020) before the citation for clarity (page 6, line 138). 

 

Reviewer #4: 

The manuscript by Cassandra J. McGill et al., titled " Sex-dimorphic expression of extracellular matrix genes in mouse bone marrow neutrophils," has been well written and study aimed to investigate the effects of innate immune system in mammals displays sex-based differences, with neutrophils, a key type of white blood cell, constituting the front-line defense. The primary focus of this study is the extracellular matrix (ECM)-related genes emerged as prominent contributors among these sex-specific genes. The author has delved into the transcriptomic patterns of primary mouse bone marrow neutrophils on both bulk and single-cell levels and found that ECM's has role in governing innate immune responses. Furthermore, this study found female-biased expression in mouse neutrophil ECM gene clusters using existing ECM gene sets and weighted gene co-expression network analysis. In summary, author found the increased prevalence of immune-related disorders like rheumatoid arthritis in women, these findings may provide light on sex-influenced inflammatory diseases. While the study provides valuable insights into the differential expression of extracellular matrix (ECM)-related genes in neutrophils based on organismal age and biological sex, there are several limitations to consider:

We thank the reviewer for recognizing the impact of our findings. We thank them for their suggestions and believe we have now addressed their stated concerns in the revised manuscript. 

Limitation:

• The study focuses primarily on transcriptomic data. While gene expression levels provide important information, they do not directly reflect protein levels or functional outcomes. Post-transcriptional and post-translational regulatory mechanisms could influence the actual ECM-related protein production and function.

• Justify the fact that in this study's author only focus on neutrophils and ECM-related genes represents a simplified view of the complex interactions occurring within the immune system and the extracellular matrix. Neutrophils are just one component of a broader cellular network, and their interactions with other immune cells are not considered under this study.

• The study focuses predominantly on gene expression and lacking functional validation of the roles of identified ECM-related genes in neutrophil biology, ECM remodeling, and disease pathogenesis.

• The study provides a descriptive analysis of gene expression patterns but lacks mechanistic insights into how these genes contribute to neutrophil function, ECM remodeling, and disease outcomes, discuss this as limitation of this study?

We now include a new paragraph in the discussion to highlight these very important limitations raised by the reviewer to contextualize our findings for readers. 

“This analysis offers new insights into the potential molecular underpinning of sex-differences in ECM biology. Our analysis, which focuses on differential expression of ECM-related gene sets, reveals a potential source for female-biased ECM production, which has not been previously explored and provides new insights for the scientific community to pursue. While our research primarily hinges on transcriptomic data, it does not offer direct insights into protein levels or functional outcomes. Future studies using targeted CRISPR or shRNA screens in neutrophil cell lines (e.g. MPRO) or use of transgenic mouse lines carrying mutations in ECM-related genes will be useful to understand the role of the clusters found in our data in setting sex-dimorphic physiological responses of neutrophils. Additionally, the intricate regulation of post-transcriptional and post-translational mechanisms may contribute substantial influence over the ECM, thus necessitating further investigation. Our study focuses on neutrophils, which are one part of the complex immune system, and our analyses do not encompass their interactions with other cells, thereby limiting the overall understanding of these intricate relationships. Ultimately, future studies investigating the mechanisms by which the ECM influences neutrophil activity and its interaction with other immune cells are essential, as well as examining how these features are conserved or differ from human neutrophils.” (page 10, line 237-252).

We thank the reviewer for helping us improve the rigor of our manuscript with this addition. 

Minor comments:

• Explain why this study hints at potential implications for managing diseases such as rheumatoid arthritis by investigating female-biased processes in neutrophils, including ECM-related factors. How can these findings be translated to clinical applications for improving therapeutic interventions in conditions involving neutrophil-mediated responses?

We agree that discussing potential implications of our findings is important for our readers.

The ECM is known to be involved in promoting these diseases and might provide a potential therapeutic handle for personalized therapies. For instance, based on the use of agonists or antagonists of sex hormone signaling. We now discuss this: “Thus, investigating female-biased processes in neutrophils, including the contribution of the ECM to promote pathology, might be an important next step in managing rheumatoid arthritis. These female-biased signatures suggest that targeting sex hormone signaling could be a potential therapeutic translation of these findings” (page 10, line 233-236).

• Discussion underscores the importance of understanding sex-specific gene expression differences in neutrophils to develop tailored therapeutic strategies. How can the insights gained from this study be leveraged to design interventions that consider both male and female-specific responses?

We believe that this question is thematically related to the above one. As noted above, we believe that targeting sex hormone signaling would be the potential next step for therapeutic translation of these findings (see new added sentence, page 10, line 233-236). 

• While the study identifies ECM-related gene clusters, functional validation is lacking. How can these genes' roles be experimentally confirmed in modulating neutrophil behavior, ECM remodeling, and disease outcomes, Need explanation?

As the reviewer likely knows, primary neutrophils are terminally differentiated and cannot be kept in vitro for genetic modifications of treatment since they are extremely short-lived (<12 hours). Thus, functional validation of findings, especially for a pure bioinformatic study, is beyond the scope of our current study.

However, to help the community’s next steps in potential validation of such findings, we have now added potential avenues for validation to our discussion: “Future studies using targeted CRISPR or shRNA screens in neutrophil cell lines (e.g. MPRO) or use of transgenic mouse lines carrying mutations in ECM-related genes will be useful to understand the role of the clusters found in our data in setting sex-dimorphic physiological responses of neutrophils.” (page 10, line 242-245).

• The study discusses the interplay between neutrophil activation, ECM deposition, and wound healing. How does neutrophil activation status affect ECM-related gene expression, and how does this interaction influence various disease contexts? Cite some study.

Neutrophil chromatin is extremely compact [PMID: 30564248], and they are known to have very little transcriptional flexibility in their mature states (with chromatin being largely impervious to transcription in the mature neutrophil configuration). In addition, they die within hours of activation. Therefore, we do not believe neutrophil activation would change ECM expression in any measurable way.

Indeed, based on supplemental figure S1A, the circulating states (G5a-c) all have similar expression of ECM genes to the progenitor states (G3-G4) [PMID: 32719519]. 

• Write the brief paragraph regarding the limitation of this study.

We thank the reviewer for suggesting this and now include a paragraph in the discussion to highlight and contextualize limitations of this study as suggested (see above). 

 

Journal Requirements:

We have ensured that our manuscript meets PLOS ONE’s style requirements including font size, figure/table labeling, and file naming. 

“Some panels created with BioRender.com. This work was supported by NIA T32 AG052374 predoctoral fellowship to C.J.M., by the Swiss National Science Foundation Funding from the SNF P3 Project 190072 to CYE, NIA R01 AG076433 to B.A.B. and Pew Biomedical Scholar award #00034120 to B.A.B.”

“This work was supported by National Institute of Aging [https://www.nia.nih.gov/] T32 AG052374 predoctoral fellowship to C.J.M., by the Swiss National Science Foundation [https://www.snf.ch/en] from the SNF P3 Project 190072 to CYE, National Institute of Aging [https://www.nia.nih.gov/] R01 AG076433 to B.A.B. and Pew Biomedical Scholar award #00034120 from the Pew Charitable Trust [https://www.pewtrusts.org/] to B.A.B. The funders had no role in study design, data collection and analysis, decision to publish, or preparation of the manuscript.”

We removed the funding information from the acknowledgements section as requested. The funding statement is correct as it currently reads and does not need further editing. 

We have revised the data availability statement in the manuscript to include URLs or accession for all necessary data files.

The Data Availability statement now reads “The bulk RNA-seq data was previously described in Lu et al., 2021 [11], and sequencing data is accessible through BioProject PRJNA630663 (https://www.ncbi.nlm.nih.gov/sra?LinkName=bioproject_sra_all&from_uid=630663). The processed normalized count table and DEseq2 result tables were obtained from Github (https://github.com/BenayounLaboratory/Neutrophil_Omics_2020). The single-cell RNA-seq data was previously described in Kim et al., 2022 [12], and raw sequencing data is accessible through BioProject PRJNA796634 (https://www.ncbi.nlm.nih.gov/sra?LinkName=bioproject_sra_all&from_uid=796634). The processed annotated Seurat file was obtained from Figshare (https://doi.org/10.6084/m9.figshare.19623978). The serum proteomics dataset was previously described in Aumailley et al, 2021 [22], and the processed protein expression matrix was obtained from the online Table S2 for the article (https://pubs.acs.org/doi/suppl/10.1021/acs.jproteome.1c00542/suppl_file/pr1c00542_si_006.xlsx)” (page 15, line 357-370). 

4. We note that Figure 1A in your submission contain copyrighted images. All PLOS content is published under the Creative Commons Attribution License (CC BY 4.0), which means that the manuscript, images, and Supporting Information files will be freely available online, and any third party is permitted to access, download, copy, distribute, and use these materials in any way, even commercially, with proper attribution. For more information, see our copyright guidelines: http://journals.plos.org/plosone/s/licenses-and-copyright.

We have obtained the CCBY permissions form from BioRender to use the graphic elements and have included the permission letter in this response package.

We checked each reference and ensured that our reference list is complete, correct, and that none of the manuscripts have been retracted.

---

## [Editor Report · Decision Letter 1]

9 Nov 2023

Sex-dimorphic expression of extracellular matrix genes in mouse bone marrow neutrophils

PONE-D-23-16855R1

Dear Dr. Benayoun,

We’re pleased to inform you that your manuscript has been judged scientifically suitable for publication and will be formally accepted for publication once it meets all outstanding technical requirements.

Kind regards,

Syed M. Faisal, Ph.D.

Academic Editor

PLOS ONE
---

## [Editor Report · Acceptance letter]

14 Nov 2023

PONE-D-23-16855R1 

Sex-dimorphic expression of extracellular matrix genes in mouse bone marrow neutrophils 

Dear Dr. Benayoun:

I'm pleased to inform you that your manuscript has been deemed suitable for publication in PLOS ONE. Congratulations! Your manuscript is now with our production department. 

Kind regards, 

on behalf of

Dr. Syed M. Faisal 

Academic Editor

PLOS ONE